# Role of Tumor-Infiltrating Lymphocytes and the Tumor Microenvironment in the Survival of Malignant Parotid Gland Tumors: A Two-Centre Retrospective Analysis of 107 Patients

**DOI:** 10.3390/jcm13123574

**Published:** 2024-06-18

**Authors:** Pietro De Luca, Arianna Di Stadio, Gerardo Petruzzi, Francesco Mazzola, Milena Fior, Luca de Campora, Matteo Simone, Pasquale Viola, Giovanni Salzano, Chiara Moscatelli, Filippo Ricciardiello, Alfonso Scarpa, Francesco Antonio Salzano, Raul Pellini, Marco Radici, Angelo Camaioni

**Affiliations:** 1Otolaryngology Department, Isola Tiberina—Gemelli Isola Hospital, 00186 Rome, Italy; marco.radici.fw@fbf-isola.it; 2Otolaryngology Department, University of Catania, 95124 Catania, Italy; arianna.distadio@unict.it; 3Department Otolaryngology Head and Neck Surgery, IRCCS Regina Elena National Cancer Institute, Istituti Fisioterapici Ospitalieri (IFO), 00144 Rome, Italy; gerardo.petruzzi@ifo.it (G.P.); francesco.mazzola@ifo.it (F.M.); milena.fior@ifo.it (M.F.); raul.pellini@ifo.it (R.P.); 4Otolaryngology Department, San Giovanni-Addolorata Hospital, 00184 Rome, Italy; ldecampora@hsangiovanni.roma.it (L.d.C.); matteo.simone@hsangiovanni.roma.it (M.S.);; 5Unit of Audiology, Department of Experimental and Clinical Medicine, Regional Centre for Cochlear Implants and ENT Diseases, Magna Graecia University, 88100 Catanzaro, Italy; pasqualeviola@unicz.it; 6Neurosciences Reproductive and Odontostomatological Sciences Department, University of Naples “Federico II”, 80138 Naples, Italy; giovanni.salzano@unina.it; 7Department of Radiological, Oncological and Pathological Sciences, Sapienza University of Rome, 00185 Rome, Italy; chiara.moscatelli@uniroma1.it; 8Otolaryngology Unit, AORN Cardarelli, 80131 Naples, Italy; filippo.ricciardiello@aocardarelli.it; 9Department of Medicine, Surgery and Dentistry, University of Salerno, 84084 Salerno, Italy; ascarpa@unisa.it (A.S.); fasalzano@unisa.it (F.A.S.)

**Keywords:** parotid cancer, major salivary gland, parotidectomy, TILs

## Abstract

**Background:** This study aims to retrospectively investigate the prognostic significance of the tumor microenvironment, with a focus on TILs (tumor-infiltrating lymphocytes), in relation to survival in a large cohort of patients with parotid gland cancer, and it uses the method proposed by the International TILs Working Group in breast cancer. **Methods:** We included a cohort of consecutive patients with biopsy-proven parotid cancer who underwent surgery between January 2010 and September 2023. A retrospective review of medical records, including surgical, pathological and follow-up reports, was performed. The density of TILs was determined according to the recommendations of the International TILs Working Group for breast cancer. **Results:** A weak negative correlation (*p* = 0.3) between TILs and time of survival and a weak positive correlation (*p* = 0.05) between TILs and months of survival (high TILs were correlated with longer survival in months) were identified. High TILs were weakly negatively, but not statistically significantly *p* (0.7), correlated with the grading of tumor; this means that high TILs were associated with low-grade tumors. **Conclusions:** Contrary to previous preliminary reports, this retrospective work found no statistically significant prognostic role of TILs in parotid gland malignancies. This case series represents the largest cohort ever reported in the literature and includes all malignant histological types. Future larger molecular studies may be useful in this regard.

## 1. Introduction

Malignant salivary gland cancers (SGCs) are rare tumors, representing only 3% of all head and neck cancers; in most cases, they are benign (80%) epithelial tumors arising from the parotid gland [1,2]. In the context of parotid tumors, mucoepidermoid carcinoma is the most common, followed by adenoid cystic carcinoma and acinic cell carcinoma [2]. These tumors present a wide and complex range of biological behaviors and are morphologically heterogeneous, making histopathological diagnosis challenging [3]. The World Health Organization (WHO) has recently updated the classification of SGCs by introducing molecular data to define new entities [4]; however, molecular typing in this field is still in progress [5]. Currently, most tumors are defined on the basis of histological and immunohistochemical findings, and molecular characterization is not mandatory for diagnosis [2,5].

Some researchers have speculated about the role of the tumor microenvironment, and in particular tumor-infiltrating lymphocytes (TILs), in the prognostic stratification of parotid gland tumors. TILs play an important role in the antigen-specific tumor immune response, and recent studies have shown that higher levels of TILs could be a negative prognostic factor for survival in SGC [6,7].

However, conclusions regarding the role of TILs in parotid gland malignancies are far from conclusive. For this reason, we want to investigate the role of TILs as a prognostic factor in malignant parotid gland tumors; our attention was focused, in particular, on survival, adjuvant treatment and metastasis. We used the method proposed by the International TILs Working Group for breast cancer, and to standardize the method of evaluation of TILs in parotid gland cancer [8]. To our knowledge, this is the largest cohort ever studied.

## 2. Materials and Methods

### 2.1. Study Design and Study Population

This multicenter retrospective study involved the Otolaryngology–Head and –Neck Surgery Departments of the San Giovanni Addolorata Hospital (Rome, Italy) and IRCCS Regina Elena National Cancer Institute (Rome, Italy). The ethical approval was waived by the local ethics committee, and due to the retrospective, non-interventional nature of the study, no number was assigned. The study was conducted in accordance with the tenets of the Declaration of Helsinki. All clinical data required for the study were recorded in a computerized database. Patients who were alive at the time of the study were informed of the study by telephone, and none refused to be included.

We consecutively and retrospectively enrolled all patients with histologically confirmed malignancy of the parotid gland who had undergone surgery between January 2010 and September 2023. We performed a review of medical records including surgical, pathological and follow-up reports. Data relating to sex, age at diagnosis, type of initial surgery (including possible neck dissection at initial surgery), clinical lymph node (cN) or distant metastasis at diagnosis, postoperative resection margins, adjuvant chemoradiotherapy (RTChT), locoregional or distant recurrence, and follow-up status (including follow-up time in months) were collected. The tumors, in all cases, were staged according to the 8th edition of the American Joint Committee on Cancer (AJCC) staging manual [9]. In the case of patients treated before 2017, histological reports were reviewed to ensure the appropriateness of the tumors’ re-staging. The data concerning survival were obtained from mortality registers, outpatient visits and radiological follow-up.

Inclusion and exclusion criteria were defined.

The inclusion criteria included an age of 18–99 years, a histological diagnosis of malignant parotid cancer according to the updated classification of SGCs by the World Health Organization [4] and availability of clinical and radiological follow-up.

The exclusion criteria included no available clinical follow-up, secondary metastatic disease to the parotid gland or tumors arising from minor salivary glands, a change in initial histological diagnosis during the review of histological slides, tumor recurrence, or the impossibility of reviewing histological slides (defects in slides and blocks or insufficient material for pathological analysis).

The data collected at each center were entered into a common Excel file, which was used to perform the statistical analyses once all the data had been entered. The data from all centers were analyzed by the first and second authors (PDL and ADS) and then shared with the other researchers to analyze the results and define the conclusions.

### 2.2. Histopathological Analysis

Each center provided the requested histological slides, which were prepared with the same protocols for fixation and pathological examination. The included pathologists had more than 15 years of experience in their field and were blinded to clinical information, treatment regimens and study endpoints. In addition, the central pathology reviewer was blinded to the interpretations of other pathologists.

The slides were reviewed independently by two pathologists: a central pathology reviewer (LC) and the institution’s local head and neck pathologist.

Based on the latest indication of WHO Classification of Head and Neck Tumors [10], the following findings were considered (i) histology of the tumor; (ii) grade of the tumor (low or high grade); (iii) postoperative resection margins (according to Hermanek and Wittekind) (R0, Rclose, R1, or R2) [11]; (iv) lymphovascular invasion (LVI) (absent, focal or <2 figures of LVI, diffuse or >2 figures of LVI); (v) perineural invasion (absent or present); (vi) extraglandular growth (absent or present).

### 2.3. Quantification of TILs

To date, despite several assays, there are no standardized systems to measure TILs yet. In 2014, an International TILs Working Group defined a set of recommendations for standardizing the evaluation of TILs in breast cancer [8]. The stromal density of TILs was determined according to the recommendations of the International TILs Working Group [8]. We selected the tumor area at low magnification and assessed the percentage of area filled with mononuclear cells in the stromal area around the tumor margin at high magnification (×200) [12]. We considered all mononuclear cells, including lymphocytes in the stromal area, to be TILs, and excluded granulocytes and other polymorphonuclear leukocytes. We followed the guidelines defined by the International TILs Working Group as a reference for evaluating the density of TILs. Examples of the evaluation are shown in Figure 1.

### 2.4. Statistical Analysis

Statistical tests were used to analyze the data. The Pearson (P) test was used to correlate TILs with grading, tumor, metastasis, overall survival (OS) and treatment used. Multivariate analyses were performed to evaluate the effect of age, tumor histotype, TILs, tumor grade (high/low), tumor size and extension (pT) and pN on status at follow-up (y); then tumor histotype, TILs, tumor grade (high/low), pT, pN effect on adjuvant therapy (y) and finally tumor histotype, TILs, tumor grade (high/low), pT and pN effect on local or distant recurrence (y). A *p*-value < 0.05 was considered statistically significant. Analyses were performed using Stata^®^ (Stata 18, StataCorp LLC.: College Station, TX, USA).

## 3. Results

### 3.1. Population of the Study, Staging and Treatment Characteristics

A total of 113 patients were diagnosed with parotid gland cancer during the study period. Patients were excluded because the histological slides could not be reviewed (n = 3) or because they were lost to follow-up (n = 3). Therefore, the study group included 107 patients with parotid malignancy. The median age was 56 years (SD 17.34, range: 15–87 years) and there was a greater proportion of female participants (F = 70, M = 37). The incidence peaked in the fifth decade (n = 28, 26%).

Pre-operative evidence of cervical node involvement (cN+) was identified in 14% (n = 15) of cases. None of the patients in this study had distant metastases at the time of their initial diagnosis of parotid cancer.

According to the European Salivary Gland Society Classification [13], most patients underwent I-II (n = 40, 37%) and I-IV (n = 49, 46%) parotidectomy; facial nerves were sacrificed in only five operations (5%). The mean surgical time was 68 min (47–123, SD 16.74).

Based on postoperative staging, we drew the following conclusions.

A total of 46 patients (43%) were pT2, 37 (35%) were pT1, 14 were pT4a (13%) and 10 (9%) were pT3.

Ninety-four patients (88%) were pN0, three (3%) were pN1, seven were pN2 (6%; n = 2 pN2a, n = 5 pN2b) and three were pN3a (3%).

ND was performed on 43 patients (40%). A total of 31 patients (29%) received adjuvant RT, while 5 patients (5%) received adjuvant RTChT; 71 patients (66%) did not receive any adjuvant treatment.

Detailed patient demographics and oncological data are summarized in Table 1.

### 3.2. Survival Outcomes

Follow-up data were available for all the patients included in this study.

At the last follow-up, 84 patients (78.5%) were alive with no evidence of disease (NED), 10 patients (9%) were alive with evidence of disease (AWD), while more patients died from other causes than from disease (DOOC; n = 8, 7% vs. DOC; n = 5, 5.5%).

With regard to recurrence rate, our findings were as follows:-Ten patients (9%) had local recurrence;-One patient (1%) had cervical lymph node recurrence;-Eight patients (7.5%) had distant metastases (mainly pulmonary).

### 3.3. Histopathological Analysis and Statistical Analysis of the Prognostic Value of TILs

According to the latest WHO classification, 42 tumors (40%) were diagnosed as mucoepidermoid carcinoma (MEC), 17 as adenoid cystic carcinoma (AdCC) (16%), 17 as acinic cell carcinoma (AciCC) (17%), 14 as carcinoma ex pleomorphic adenoma (CExPA) (13%), 11 as salivary duct carcinoma (SDC) (10%), 3 as polymorphous adenocarcinoma (PAC) (3%) and 1 each as squamous cell carcinoma (SCC) (0.5%), lymphoepithelial carcinoma (LEC) (0.5%) and oncocytic carcinoma (OC) (0.5%).

The majority of tumors showed no LVI (n = 84, 78% versus n = 23, 22%), no tumor necrosis (n = 76, 71% versus n = 31, 29%), no PNI (n = 75, 70% versus n = 32, 30%), no extra glandular growth (n = 65, 61% versus n = 42, 39%), and clear postoperative margins (R0, n = 62, 58%). Seventy-three patients (68%) had a low-grade tumor, while thirty-four patients (32%) had a high-grade tumor.

According to the score proposed by Salgado et al. [8], there were more tumors with a low score (n = 53, 50%) and a moderate score (n = 20, 18.5%) than those with an intense score (n = 14, 13%); TILs were absent in only 20 cases (18.5%). Patients’ detailed histological data are summarized in Table 2.

We identified a weak negative correlation, though it was not statistically significant (R = −0.1026; *p* = 0.3), between high TILs and survival without disease (NED); this indicated that high TILs were correlated with a better outcome (NED).

A weak positive, statistically insignificant correlation (R = 0.1024; *p* = 0.3) was identified between high TILs and months of survival; the high TILs were correlated with longer OS in months.

A weak negative, statistically insignificant correlation (R = −0.029; *p* = 0.7) was identified between high TILs and treatment; high TILs were correlated with the absence of use of adjuvant treatments (chemo or radiotherapy), so the patients who had high TILs only needed surgery.

High TILs were weakly negatively correlated, despite not being statistically significant *p* (R = −0.0031; *p* = 0.9), with the grading of the tumor; this means that high TILs were associated with low-grade tumors.

High TILs were weakly positively correlated, despite not being statistically significant *p* (R = 0.0389; *p* = 0.6), with pT; this means that high TILs were associated with tumors of small dimensions (<2 cm) and without extraparenchymal involvement (T1).

Finally, high TILs were weakly negatively correlated, without being statistically significant *p* (R = −0.0011; *p* = 0.9), with pN; in the presence of high TILs, the patients had pN0.

#### Results Multilinear Regression Analyses

State at follow-up. All variables (age, tumor histotype, TILs, tumor grade (high/low), p and pN) were weakly correlated with the state at the follow-up (intercept: 0.47). However, age alone had a statistically significant effect (*p* = 0.001).

Adjuvant therapy. All variables (tumor histotype, TILs, tumor grade (high/low), pT and pN) were weakly correlated with the state at follow-up (intercept: 0.54). However, only at intercept did we identify a statistically significant effect (*p* = 0.01) (Figure 2).

Local/distant recurrence. All variables (tumor histotype, TILs, tumor grade (high/low), pT and pN) were correlated with the state at follow-up (intercept: −0.47; *p* < 0.0001). We identified statistically significant effects at intercept (*p* = 0.001) and the following variables were used to determine the tumor grade (*p* = 0.004) and pT (*p* < 0.0001) (Figure 3).

## 4. Discussion

Contrary to previous preliminary reports, this retrospective work identified a potential protective role of TILs in parotid gland malignancies, despite none of the statistical tests showing a statistically significant *p* value. The absence of a statistically significant value can be attributed to the small sample size, which, despite being the largest cohort ever reported in the literature and including all malignant histological types, was nevertheless not big enough to produce statistically significant results because of the presence of many variables.

The multilinear regression analyses showed that age is a positive prognostic factor, and this is in line with what has been observed in surgical studies focused on immune senescence [14]; the immune response tends to decrease with age, and TILs do the same.

In addition, the analyses confirmed that tumor grade, dimension and invasion are fundamental elements in causing tumors to spread.

TILs are white blood cells (including T and B cells) that have left the bloodstream and migrated to a tumor, and they can be found in the stroma of the tumor [15]. One of the main roles of TILs is to kill tumor cells, so their presence and high density is often associated with better clinical outcomes (OS and response to immunotherapy) [16]. Previously, other authors have found that a high density of stromal TILs is associated with a survival benefit in breast [8] and colorectal cancer [17].

Although many studies have focused on the specific role of TILS and lymphocyte subtypes, there is no standardized protocol for the evaluation of TILs in parotid cancer that is generally accepted by pathologists. In the present study, to reduce the number of methodological errors and to standardize the evaluation of TILs between the participating centers, we proposed using the recommendation for evaluation of TILs in breast cancer by Salgado et al. [8]. According to these guidelines, the pathologist must select the tumor area and define the stromal area, then scan at low magnification, identify the inflammatory infiltrate and assess the percentage of stomal TILs (according to three percentage ranges; 0–10%, 20–40%, 50–90%) [8].

The interest in the role of the tumor microenvironment in head and neck cancers is growing widely; this is probably related to the increased need to find molecular prognostic components to improve the TNM classification in the near future, which focuses on the tumor itself and not on the extra-epithelial components.

Few studies have evaluated the value of TILs in SGC.

The prognostic value of TILs in parotid AciCC was investigated by Hiss et al., who retrospectively evaluated the correlation with clinicopathological features in a cohort of 36 primary AciCCs with long-term clinical follow-up. The authors concluded that increased immune cell infiltration of T and B cells and high levels of PD-L1 expression in AciCCs, in association with high-grade transformation, lymph node metastasis and poor prognosis, could suggest a relevant interaction between tumor cells and immune cell infiltrates. They speculated about a potential rationale for such immune checkpoint inhibition [7]. In contrast, an Italian multicenter study conducted on the largest cohort of patients with AciCC of the parotid gland did not identify any prognostic value (lymph node metastases, tumor grade, OS) of TILs [4]. Almeida de Arruda et al. identified low immunogenicity in the tumor microenvironment of a specific subtype of SGC, adenoid cystic carcinoma (AdCC) [18].

Recently, De Virgilio et al. attempted to define the role of TILs and tumor-associated macrophages (TAMs) as predictors of lymph node metastasis in major SGC. In this cohort, the primary tumor site was the parotid gland in 20 patients, the submandibular gland in 4 patients and the sublingual gland in 1 patient; to this end, a selected number of immunohistochemical markers associated with TILs (CD3, CD4, CD68 and FOXP3) and TAMs (CD68 and CD163) were examined, suggesting that the higher likelihood of LNM in the SGC may be correlated with a high density of specific TIL and TAM subpopulations [19].

Furthermore, the growing role of immunotherapy in patients with recurrent/metastatic/non-surgically treated LSCC represents a promising new treatment pathway, but there is also a need to achieve a proper characterization of immune status to understand and stratify those patients who may benefit from immune modulation. The results of our study may encourage observational studies to evaluate the association between TILs density and the molecular assets of parotid cancers.

### Limitations of the Study

Several limitations should be considered when assessing the significance and generalizability of these findings. This was a retrospective analysis, and data on treatment decisions were not available.

The number of patients included may not be satisfactory for a general oncology cohort; however, considering the rarity of parotid gland malignancies, our cohort can be considered extremely representative.

Although there is no standardized system for evaluating TILs in H and N malignancies, we used the score proposed by the International TILs Working Group in breast cancer and attempted to standardize the method for evaluating TILs in parotid cancer.

Furthermore, given the exploratory nature of our study and the cost of the pathological procedures, we decided not to characterize the type of circulating lymphocytes; however, given the results obtained, lymphocyte typing will be the aim of the definitive study.

## 5. Conclusions

This retrospective work, which represents the largest cohort ever reported in the literature and includes all malignant histological types, did not find a statistically significant prognostic role of TILs in parotid gland malignancies. Notwithstanding the results of this work, stratifying the different types of TILs in relation to individual histological variants could represent an attempt to understand whether TILs play a prognostic role in specific subtypes of parotid malignancies. Larger molecular studies in the future may be useful in this regard.

## Figures and Tables

**Figure 1 jcm-13-03574-f001:**
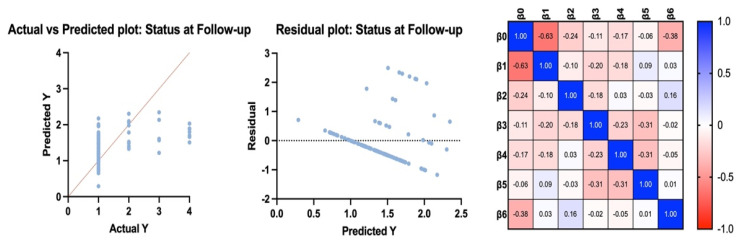
Results of multilinear regression considering the Y state at follow-up.

**Figure 2 jcm-13-03574-f002:**
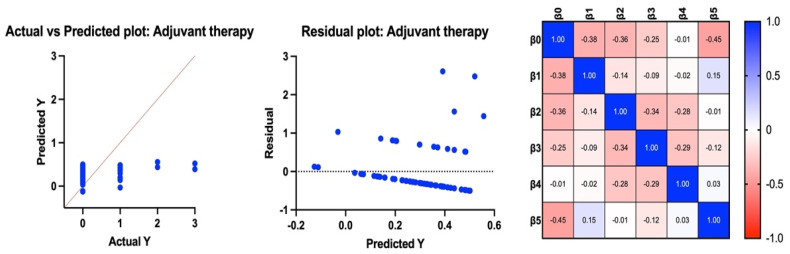
Results of multilinear regression considering Y to be an indicator of a need for adjuvant therapy.

**Figure 3 jcm-13-03574-f003:**
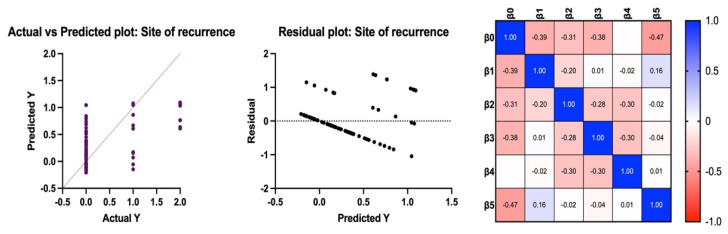
Results of multilinear regression considering Y as an indicator of local or distant recurrence/metastasis.

**Table 1 jcm-13-03574-t001:** Cohort of patients with parotid cancers: epidemiological, clinical and survival characteristics.

*Variable*		N
** *Participants* **	107	
** *Mean age (min-max; SD)* **	56 (15–87; 17.34)	
** *Gender* **	Male	37
	Female	70
** *cN+* **		15
** *Type of parotid surgery* **	I	
	I-II	40
	I-II-III	
	I-IV	49
	I-V	5
** *Neck dissection (ND)* **	Ipsilateral	43
** *pT staging* **	pT1	37
	pT2	46
	pT3	10
	pT4a	14
** *pN staging* **	pN0	94
	pN1	3
	pN2	7
	pN3	3
** *Adjuvant treatment* **	RT	31
	RTChT	5
** *Status at last follow-up* **	NED	84
	AWD	10
	DOD	5
	DOOC	8
** *Recurrence* **	Local	10
	Cervical lymph nodes	1
	Distant metastasis	8

SD—standard deviation. ND—neck dissection. RT—radiotherapy. RTChT—radiochemotherapy. NED—no evidence of disease. AWD—alive with disease. DOD—died of disease. DOOC—died of other cause.

**Table 2 jcm-13-03574-t002:** Histological features of patients with parotid cancers.

Variable		N
** *Histological features* **		
***Histological type***	MEC	42
	AdCC	17
	AciCC	17
	CExPA	14
	SDC	11
	SCC	3
	LEC	1
	PAC	1
	OC	1
***Grade***	Low	73
	High	34
***Lymphovascular invasion***	Yes	84
	No	23
***Perineural invasion***	Yes	75
	No	32
***Tumor necrosis***	Yes	76
	No	31
***Extraglandular growth***	Yes	65
	No	42
** *Postoperative margins* **	R0	62
** *TILs* **	Absent	20
	Low score	53
	Moderate	20
	Intense score	14

MEC—mucoepidermoid carcinoma. AdCC—adenoid cystic carcinoma. AciCC—acinic cell carcinoma. CExPA—carcinoma ex plemorphous adenoma. SDC—salivary duct carcinoma. SCC—squamous cell carcinoma. LEC—lymphoepitelial carcinoma. Polymorphous adenocarcinoma. OC—oncocytic carcinoma. TILs—tumor-infiltrating lymphocytes.

## Data Availability

No new data were created or analyzed in this study. Data sharing is not applicable to this article.

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
