# Peer review of "Role of Tumor-Infiltrating Lymphocytes and the Tumor Microenvironment in the Survival of Malignant Parotid Gland Tumors: A Two-Centre Retrospective Analysis of 107 Patients"

_jcm, 2024, doi:10.3390/jcm13123574_

Round 1
Reviewer 1 Report
Comments and Suggestions for Authors
The manuscript provides insights into the role of TILs in the survival of malignant parotid gland tumors.
I could not access and evaluate any figures.
Ensure the manuscript consistently states whether TILs are associated with good or adverse prognosis and provide mechanisms and study findings. Provide a clear hypothesis about the role of TILs (adverse or good prognosis). Expand on the background of TILs in the context of tumor microenvironment and prognosis.
Redesign tables for clarity and professionalism. Include abbreviations under each table.
Include figures for correlation analyses to visually represent the data.
Re-evaluate the correlation analyses to ensure accuracy. For instance, if high TILs correlate with low T, ensure both show a negative correlation instead of positive.
Discuss the mechanisms by which TILs influence prognosis regarding the study findings and existing literature.
Report the correlation analysis among pathologists to demonstrate the consistency of TIL evaluations.
Comments on the Quality of English LanguageModerate editing of English language required
Author Response
Dear Reviewers,
Thanks for reviewing our manuscript and for the valuable comments that helped us clarify some relevant aspects that were missed or unclear in the first version of the paper.
We have read carefully your comments and made the changes to address comments and concerns using track-changes.
Our answers to your concerns are blue bolded. The paper has been entirely revised to reduce plagiarism and to improve the English quality and readability.
We hope that the changes made in the revised manuscript and responses provided below have adequately addressed the reviewer’s comments and made this paper stronger.
Review 1
The manuscript provides insights into the role of TILs in the survival of malignant parotid gland tumors.
- I could not access and evaluate any figures.
We hope that now the figures are fully accessible.
- Ensure the manuscript consistently states whether TILs are associated with good or adverse prognosis and provide mechanisms and study findings. Provide a clear hypothesis about the role of TILs (adverse or good prognosis). Expand on the background of TILs in the context of tumor microenvironment and prognosis.
Thanks for your comment. We discussed this point and we also added info about the high/low TILs and their role in the result sections and in the discussion.
- Redesign tables for clarity and professionalism. Include abbreviations under each table.
Thanks for your comment. We redesigned the tables and included abbreviations.
- Include figures for correlation analyses to visually represent the data.
We included figures 1, 2 and 3 that shows the results of multilinear regression analyses.
- Re-evaluate the correlation analyses to ensure accuracy. For instance, if high TILs correlate with low T, ensure both show a negative correlation instead of positive.
We performed additional multinear regression analyses and we fully described in the material and method section (statistical analyses) which were “x” and “y” parameters. We also clarified the statistic; in particular we only used Pearson. In the results are now reported the “R” value in addition to p for clarity. We also clarified in the results the meaning of “negative” or “positive” correlation with a sentence in addition to the statistic values.
Reviewer 2 Report
Comments and Suggestions for Authors
1. Lines 120-121. “Spearman (S) and Pearson (P) tests were used to correlate TILs with grading, with tumor, with metastasis, with survival and treatment used.” Why did the authors use these tests for survival? Usually survival analyses, as time-to-event data, are specifically analyzed with log-rank test or multivariate models.
2. Survival. When survival is mentioned, is it overall survival or disease-free survival? It may be useful to specify it.
3. Where TILs have been counted? Only in the stroma, as in breast carcinoma? Or inside the tumor? It is explained in the Discussion, but a little explanation could also be provided in the M&M section.
4. Lines 174-191. P is the statistical p or is it Pearson coefficient? When there is a positive/negative correlation, a numeric value of this correlation (not only its statistical significance) could be added.
5. Why is it necessary to underline that all lesions were biopsy-proved in surgical specimens? Sometimes malignancies are discovered directly in surgical specimens. Would the authors probably underline that all lesions were histologically confirmed in surgical specimens?
Author Response
Dear Reviewers,
Thanks for reviewing our manuscript and for the valuable comments that helped us clarify some relevant aspects that were missed or unclear in the first version of the paper.
We have read carefully your comments and made the changes to address comments and concerns using track-changes.
Our answers to your concerns are blue bolded. The paper has been entirely revised to reduce plagiarism and to improve the English quality and readability.
We hope that the changes made in the revised manuscript and responses provided below have adequately addressed the reviewer’s comments and made this paper stronger.
- Lines 120-121. “Spearman (S) and Pearson (P) tests were used to correlate TILs with grading, with tumor, with metastasis, with survival and treatment used.” Why did the authors use these tests for survival? Usually survival analyses, as time-to-event data, are specifically analyzed with log-rank test or multivariate models.
Thank you for this comment. We performed multilinear regression analyses. Details about the method have been described in the material and methods (statistical analyses) and we also added three figures to visually show the results of the analyses.
2. Survival. When survival is mentioned, is it overall survival or disease-free survival? It may be useful to specify it.
Thanks for your comment. We have corrected it (OS).
3. Where TILs have been counted? Only in the stroma, as in breast carcinoma? Or inside the tumor? It is explained in the Discussion, but a little explanation could also be provided in the M&M section.
Thanks for your comment; we added the information in the methods section.
4. Lines 174-191. P is the statistical p or is it Pearson coefficient? When there is a positive/negative correlation, a numeric value of this correlation (not only its statistical significance) could be added.
We added “R” value in addition to p. Thank you for this note.
5. Why is it necessary to underline that all lesions were biopsy-proved in surgical specimens? Sometimes malignancies are discovered directly in surgical specimens. Would the authors probably underline that all lesions were histologically confirmed in surgical specimens?
Thanks for your comment; we corrected it.
Reviewer 3 Report
Comments and Suggestions for Authors
hello
thank you for the paper, its quite interesting
title =ok
abstract- explain TIL in the beginning of the abstract
abstract is structured and well presented
abstract ok
key words -ok
introduction:
introduction is short and well written
please add the list of most common parotid gland cancers in the introduction + references
please add in the introduction why diagnostics and treatment of parotid cancers is troublesome, and how potential TIL can help or improve future studies
material and methods section
good written -ok
material and methods are sound, well descried
please highlight the inclusion criteria for the study
what was the histopathological protocol consisted of? please explain
statistics are ok
results =OK
results need more clarification, how could they improve the parotid gland cancers outcomes and diagnostics
table 1 is too big and not descriptive, missing abbreviations legends - please improve
also a figure, a flow-chart of the included/excluded patients for the study should be added, and described why did it happen
surgical results are short
chapter 3.3 is well written, nothing to add more
table 2- missing abbreviations, table is too big and not descriptive enough
at the end of the result please highlight the most important results and their possible usage for future studies
discussion - is well written and sound
discussion is short, please improve it and add more TIL suggested role in parotid cancer
rest OK
study limitations are well presented
conclusions are sound
study references are suitable
rest is quite alright
please improve the paper for any future considerations
Author Response
hello, thank you for the paper, its quite interesting
title =ok
abstract- explain TIL in the beginning of the abstract
Thanks for your comment, we have corrected it.
abstract is structured and well presented. abstract ok. key words -ok
introduction:
introduction is short and well written
please add the list of most common parotid gland cancers in the introduction + references
Thanks for your comment, we added in the introduction the most common parotid malignancies.
please add in the introduction why diagnostics and treatment of parotid cancers is troublesome, and how potential TIL can help or improve future studies
Thanks for your comment, we added an explanation.
material and methods section
good written -ok
material and methods are sound, well descried
please highlight the inclusion criteria for the study
Thanks for your comment, we have included the section “inclusion criteria”.
what was the histopathological protocol consisted of?
The histopathological analysis of TILs is already explained in the methods section.
statistics are ok
results =OK
results need more clarification, how could they improve the parotid gland cancers outcomes and diagnostics
Thanks for your comment, we improved the “results” section and the statistical analysis.
table 1 is too big and not descriptive, missing abbreviations legends - please improve
also a figure, a flow-chart of the included/excluded patients for the study should be added, and described why did it happen
Thanks for these two comments. We reduced the table 1 and we added the abbreviations.
surgical results are short
Thanks for your comment; we have entered the additional operating data that we have available, i.e. the operating time.
chapter 3.3 is well written, nothing to add more
table 2- missing abbreviations, table is too big and not descriptive enough
Thank for your comment. We reduced the table’s dimensions.
at the end of the result please highlight the most important results and their possible usage for future studies
Thanks for your comment, we have added it.
discussion - is well written and sound
discussion is short, please improve it and add more TIL suggested role in parotid cancer
Thanks for your comment, we improved the discussion section.
rest OK
study limitations are well presented
conclusions are sound
study references are suitable
rest is quite alright
please improve the paper for any future considerations
We really appreciated your careful and thoughtful evaluation of our manuscript and hope that this revised version meets with your approval. We have tracked all changes using colored words in the revised manuscript. Thanks again for your interest in our work. We await your review of our revised manuscript.
Sincerely yours,
The Authors
Round 2
Reviewer 1 Report
Comments and Suggestions for Authors
The authors have addressed all my concerns
Reviewer 3 Report
Comments and Suggestions for Authors
hello
thank you for your valuable comments
with kind regards